

# Inhibition of SARS-CoV-2 infection in human iPSC-derived cardiomyocytes by targeting the Sigma-1 receptor disrupts cytoarchitecture and beating

José Alexandre Salerno[1,2,*], Thayana Torquato[2,*], Jairo R. Temerozo[3,4], Livia Goto-Silva[2], Karina Karmirian[1,2], Mayara A. Mendes[2], Carolina Q. Sacramento[5,6], Natalia Fintelman-Rodrigues[5,6], Letícia R Q. Souza[2], Isis M. Ornelas[2], Carla P. Veríssimo[1], Luiz Guilherme H S. Aragão[2], Gabriela Vitória[2], Carolina S G. Pedrosa[2], Suelen da Silva Gomes Dias[5], Vinicius Cardoso Soares[1,5], Teresa Puig-Pijuan[2,7], Vinícius Salazar[8], Rafael Dariolli[9,10], Diogo Biagi[10], Daniel R. Furtado[2], Luciana Barreto Chiarini[7], Helena L. Borges[1], Patrícia T. Bozza[5], Marilia Zaluar P. Guimarães[1,2], Thiago M.L. Souza[5,6] and Stevens K. Rehen[2,11]

[1] Institute of Biomedical Sciences, Federal University of Rio de Janeiro (UFRJ), Rio de Janeiro, Brazil
[2] D'Or Institute for Research and Education (IDOR), Rio de Janeiro, Brazil
[3] National Institute for Science and Technology on Neuroimmunomodulation (INCT/NIM), Oswaldo Cruz Institute (IOC), Oswaldo Cruz Foundation (Fiocruz), Rio de Janeiro, Brazil
[4] Laboratory on Thymus Research, Oswaldo Cruz Institute (IOC), Oswaldo Cruz Foundation (Fiocruz), Rio de Janeiro, Brazil
[5] Immunopharmacology Laboratory, Oswaldo Cruz Institute (IOC), Oswaldo Cruz Foundation (Fiocruz), Rio de Janeiro, Brazil
[6] National Institute for Science and Technology on Innovation in Diseases of Neglected Populations (INCT/IDPN), Center for Technological Development in Health (CDTS), Oswaldo Cruz Foundation (Fiocruz), Rio de Janeiro, Brazil
[7] Carlos Chagas Filho Institute of Biophysics (IBCCF), Federal University of Rio de Janeiro (UFRJ), Rio de Janeiro, Brazil
[8] Department of Systems and Computer Engineering, COPPE, Federal University of Rio de Janeiro (UFRJ), Rio de Janeiro, Brazil
[9] Department of Pharmacological Sciences, Icahn School of Medicine at Mount Sinai, New York, NY, United States of America
[10] PluriCell Biotech, São Paulo, Brazil
[11] Department of Genetics, Institute of Biology, Universidade Federal do Rio de Janeiro, Rio de Janeiro, Brazil
[*] These authors contributed equally to this work.

Corresponding author
Stevens K. Rehen, srehen@lance-ufrj.org

## ABSTRACT

SARS-CoV-2 infects cardiac cells and causes heart dysfunction. Conditions such as myocarditis and arrhythmia have been reported in COVID-19 patients. The Sigma-1 receptor (S1R) is a ubiquitously expressed chaperone that plays a central role in cardiomyocyte function. S1R has been proposed as a therapeutic target because it may affect SARS-CoV-2 replication; however, the impact of the inhibition of S1R in human cardiomyocytes remains to be described. In this study, we investigated the consequences of S1R inhibition in iPSC-derived human cardiomyocytes (hiPSC-CM). SARS-CoV-2 infection in hiPSC-CM was productive and reduced cell survival. S1R inhibition decreased both the number of infected cells and viral particles after 48 hours. S1R inhibition also prevented the release of pro-inflammatory cytokines and cell death.

Although the S1R antagonist NE-100 triggered those protective effects, it compromised cytoskeleton integrity by downregulating the expression of structural-related genes and reducing beating frequency. Our findings suggest that the detrimental effects of S1R inhibition in human cardiomyocytes' integrity may abrogate its therapeutic potential against COVID and should be carefully considered.

## INTRODUCTION

COVID-19 is an airborne infectious disease caused by the Severe Acute Respiratory Syndrome Coronavirus 2 (SARS-CoV-2). Since the first patients were diagnosed, myocardial injury following SARS-CoV-2 infection has been reported increasing the in-hospital mortality (of 51.2%) in comparison to cases without cardiac injury (of 4.5%) (*Huang et al., 2020*; *Gnecchi et al., 2020*; *Shi et al., 2020*). Pre-existing cardiovascular comorbidities are related to worse clinical outcomes and, together with diabetes, are the most common chronic conditions among hospitalized COVID-19 patients (*Yang et al., 2020*; *Magadum & Kishore, 2020*). Hence, there is a higher prevalence of cardiovascular complications as extrapulmonary COVID-19. Together, these data reinforce that therapeutic approaches targeting SARS-CoV-2 infection must not only seek to diminish cardiac damage by COVID-19, but also consider the eventual drug-induced damage to the heart, which would in turn aggravate the disease (*Pellicori et al., 2021*).

Human induced pluripotent stem cell-derived cardiomyocytes (hiPSC-CMs) reproduce key features of human myocardial cells, such as expression of lineage-specific markers, sarcomeric and cytoskeletal organization, subcellular structures, and contractility (*Karakikes et al., 2015*). Given the scarcity of cardiomyocytes from other sources and the suitability of hiPSC-CMs as a model *in vitro* for cardiac diseases, they are useful tools to investigate SARS-CoV-2 infection and to test drugs that may eventually prevent cardiac susceptibility. Indeed they were used for these purposes and it was shown that the new coronavirus infects these cells and causes mechanical and electrical impairment (*Sharma et al., 2020*; *Choi et al., 2020*; *Wong et al., 2020*; *Perez-Bermejo et al., 2021*; *Marchiano et al., 2021*). Also, these cells are reliable to evaluate drug-induced cardiotoxicity, which is one of the main causes of drug withdrawal from the market due to fatal side effects (*Chaudhari et al., 2016*; *Sharma et al., 2018*; *Choi et al., 2020*). hiPSC-CMs can be used as a preclinical model to investigate the safety of emerging therapeutics and more effective treatments for COVID-19.

Compounds from drug repurposing studies have been proposed to display antiviral activity against SARS-CoV-2 (*Lovato et al., 2020*). Negative modulators of the Sigma-1 receptor (S1R) interact with the Non-Structural Protein 6 (NSP6) from SARS-CoV-2 (*Gordon et al., 2020b*; *Hashimoto, 2021*). Knockout of S1R decreased the replication of SARS-CoV-2 in Caco-2 cells (*Gordon et al., 2020a*). The NSP6 is one of the proteins assembled into the replication complex built in the endoplasmic reticulum (ER) of host

cells infected by other coronaviruses (*Cottam et al., 2011a*; *Cottam, Whelband & Wileman, 2014*). The role of NSP6 in the intracellular viral cycle has been associated with the formation of vesicles in sites of replication and autophagosomes to favor the replication in the ER and ER-derived vesicles (*Cottam et al., 2011b*; *Cottam, Whelband & Wileman, 2014*). Interestingly, this is the subcellular localization where S1R is enriched, especially within the ER-mitochondrion contact (*Dussossoy et al., 1999*; *Hayashi & Su, 2003a*; *Hayashi & Su, 2003b*; *Hayashi & Su, 2007*; *Su et al., 2010*; *Brune, Pricl & Wünsch, 2013*; *Rousseaux & Greene, 2016*; *Gordon et al., 2020b*). However, this receptor is an ER chaperone that acts assisting in the folding of host proteins, either during their synthesis or function, and controlling calcium homeostasis (*Morales-Lázaro, González-Ramírez & Rosenbaum, 2019*). S1R is also a key intracellular amplifier of signal transduction in different pathways including bioenergetics and response to cellular stress (*Hayashi & Su, 2007*; *Hayashi et al., 2009*; *Hayashi, 2015*). Therefore, although targeting this receptor was shown to block SARS-CoV-2 replication, it may also lead to harmful side effects because of its involvement in vital cellular functions.

In the context of cardiac tissue, S1R plays important roles in cardioprotection against maladaptive ER stress responses, apoptosis after ischemic injuries and hypertrophy (*Tagashira & Fukunaga, 2012*; *Tagashira et al., 2013*; *Tagashira et al., 2014*; *Hirano, Tagashira & Fukunaga, 2014*; *Lewis et al., 2020*). Research both *in vivo* and *in vitro* have proposed calcium homeostasis and suppression of ER stress induced toxicity as possible events related to S1R cardiovascular function in rodents (*Bhuiyan & Fukunaga, 2009*; *Tagashira et al., 2013*; *Shinoda et al., 2016a*; *Qin et al., 2019*). Accordingly, depletion of S1R has been demonstrated to cause mitochondrial damage and contractile dysfunction (*Abdullah et al., 2018*; *Abdullah et al., 2020*).

Drugs that bind to Sigma receptors, even as an off-target, have been considered as potential therapeutics to prevent or to treat COVID-19 (*Reznikov et al., 2020*; *Vela, 2020*). Some of these compounds have been reported to exhibit antiviral activity against other coronaviruses, together with cardiotoxicity and induction of arrhythmias (*Chen, Wang & Lin, 2006*; *Page et al., 2016*). Others, however, have stated that the antiviral properties of these compounds are due to their physicochemical properties rather than their pharmacological interaction with S1R (*Tummino et al., 2021*). Therefore, the modulation of S1R is in the spotlight of alternative approaches to treat COVID-19 but it remains to be explored in the perspective of efficiency and safety in human heart cells. In this study, we investigated the consequences of the inhibition of S1R in hiPSC-CMs. Inhibition of S1R led to transcriptional modifications of myofibril-associated genes, aberrant changes in the cytoskeleton and decreased beating frequency, despite reducing infection and viral replication of SARS-CoV-2 and preventing viral-associated cytokine release and cell death. These results provide new insights about targeting S1R as a strategy against COVID-19 and possible adverse cardiac outcomes.

## MATERIALS & METHODS

### hiPSC-CMs culture

Fresh human iPSCs-derived cardiomyocytes were purchased from Pluricell (São Paulo, Brazil) and the protocol for cardiomyocyte differentiation is described (*Cruvinel et al., 2020*). Upon arrival, cardiomyocytes were allowed to regain contractility and maintained at 37 °C in a humidified atmosphere with 5% $CO_2$. Cardiomyocytes were used between days 30 to 40 of differentiation.

### Chemicals

4-Methoxy-3-(2-phenylethoxy)-N,N-dipropylbenzeneethanamine hydrochloride (NE-100 hydrochloride) was purchased from Tocris (3133). Stock and work solutions were prepared using 100% dimethyl sulfoxide sterile-filtered (DMSO; D2650 - Sigma-Aldrich).

### Flow cytometry

Cardiomyocytes were plated on 6-well plates coated with GELTREX and cultivated for 7 days. After cell dissociation, cells were fixed with 1% paraformaldehyde (PFA), permeabilized with Triton 0.1% (Sigma Aldrich) and Saponin 0.1% (Sigma Aldrich) and stained with the antibodies anti-TNNT2 (1:2500; Thermo Fisher, MA5-12960) and anti-OCT4 (1:200, Thermo Fisher, MA5-14845). FC data was acquired using a Canto BD flow cytometer for each batch of differentiation and analyzed using the FlowJo Software considering 1%–2% of false-positive events.

### SARS-CoV-2 propagation

SARS-CoV-2 was expanded in Vero E6 cells from an isolate obtained from a nasopharyngeal swab of a confirmed case in Rio de Janeiro, Brazil (GenBank accession no. MT710714). The National Review Board approved the study protocol (CONEP 30650420.4.1001.0008) for clinical samples, and informed consent was obtained from all participants or patients' representatives. Viral isolation was performed after a single passage in cell culture in 150 cm² flasks with high glucose DMEM 2% FBS. Observations for cytopathic effects were performed daily and peaked 4 to 5 days after infection. All procedures related to virus culture were handled in biosafety level 3 (BSL3) multi-user facilities according to WHO guidelines. Virus titers were determined as plaque-forming units (PFU), and virus stocks were kept in −80 °C ultra-low temperature freezers.

### Infections and virus titration

Cardiomyocytes were infected with SARS-CoV-2 at MOI of 0.1 in CDM3 media without serum. After 1 h, cells were washed and incubated with complete medium with treatments or not. For virus titration, monolayers of Vero E6 cells ($2 \times 10^4$ cells/well) in 96-well plates were infected with serial dilutions of supernatants containing SARS-CoV-2 for 1 h at 37 °C. A semi-solid high glucose DMEM medium containing 2% FSB and 2.4% carboxymethylcellulose was added and cultures were incubated for 3 days at 37 °C. Then, cells were fixed with 10% formalin for 2 h at room temperature. The cell monolayer was stained with 0.04% solution of crystal violet in 20% ethanol for 1 h. Plaque numbers

were scored in at least three replicates per dilution by independent readers blinded to the experimental group and the virus titers were determined by PFU per milliliter (PFU/ml).

## Immunocytochemistry and fluorescence image analysis

Cells were fixed using 4% PFA solution (Sigma-Aldrich) for 1 h, washed with 1X PBS and then incubated with permeabilization/blocking solution (0.3% Triton X-100/3% bovine serum albumin) for 1 h. Primary antibodies were diluted in blocking solution and incubated at 4 °C overnight, namely anti-SARS-CoV-2 convalescent serum (1:1000); anti-SARS-CoV-2 spike protein monoclonal antibody (SP) (1:500, G632604; Genetex); anti-cardiac troponin T (cTnT) (1:500, MA5-12960; Invitrogen), anti-VDAC1/Porin (1:100, ab34726; Abcam), anti-Calnexin (1:50, #2433; Cell Signaling Technology) and anti-Sigma1R B-5 (1:100, SC-137075; Santa Cruz Biotechnology). The use of the convalescent serum from COVID-19 patients was approved by CAAE number: 30650420.4.1001.0008. Next, hiPSC-CMs were washed with PBS 1X and incubated with the secondary antibodies diluted in blocking solution: goat anti-Human Alexa Fluor 647 (1:400; A-21445; Invitrogen), goat anti-Rabbit Alexa Fluor 546 and goat anti-Mouse Alexa Fluor 488 (1:400; A-11001; Invitrogen) for 1 h at room temperature. Actin filaments were stained with Alexa Fluor 568 phalloidin (1:10; A-12380; Life Technologies) for 1 h. Nuclei were stained with 300 nM 4′-6-diamino-2-phenylindole (DAPI) for 5 min and each well was mounted with 50% PBS-Glycerol.

For quantitative analysis, images were acquired using Operetta® High-Content Screening System (Perkin Elmer) with a 20x long working distance (WD) objective lens from at least 10 fields per well. For cell surface area measurement, images of F-actin stained cardiomyocytes were segmented using Cellpose and the area was measured using NIH ImageJ software (*Stringer et al., 2021*). For the other analyses, data were evaluated using the Columbus™ Image Data Storage and Analysis System (Perkin Elmer) for image segmentation and object detection. Fluorescence threshold was set to determine positive and negative cells for each marker. Representative immunostaining images were acquired on a Leica TCS-SP8 confocal microscope using a 63x oil-immersion objective lens.

## Measurements of cell death and cytokines

Monolayers of hiPSC-CMs in 96-well plates (70–90% confluence) were treated with various concentrations of NE-100. The neutral red assay was performed, and viability was estimated by the percentage relative to untreated condition (vehicle) using the mean of 6 technical replicates per experiment.

The levels of IL-6, IL-8, and TNF-α were quantified in the supernatants from uninfected and SARS-CoV-2-infected hiPSC-CMs by ELISA (R&D Systems), following manufacturer's instructions. Control groups (mock and cells infected with SARS-CoV-2 only) were also analyzed in *Aragão et al. (2021)*. The results were obtained as picograms per milliliter (pg/ml) and are expressed as fold-change relative to untreated uninfected control. Cell death was determined according to the activity of lactate dehydrogenase (LDH) in the culture supernatants using a CytoTox® Kit (Promega, Madison, WI, USA) according to the manufacturer's instructions. Supernatants were centrifuged at 5,000 rpm for 1 min, to remove cellular debris.

## Gene expression analysis

Total RNA was isolated using TRIzol, according to the manufacturer's recommendations (Thermo Fisher Scientific) and digested using DNase I, Amplification Grade (Invitrogen), following the manufacturer's instructions. DNase-treated RNA samples (1 μg) were converted to complementary DNA (cDNA) using the M-MLV enzyme (Thermo Fisher Scientific). Qualitative endpoint PCR reactions were executed with the following primer sequences: S1R (forward 5′- AGTAGGACCATGCACTCACACC-3′; reverse: 5′- CCCCATCCTTAACTCTAGAACC-3′). Glyceraldehyde-3-phosphate Dehydrogenase (GAPDH; forward: 5′-TTCGACAGTCAGCCGCATC-3′; reverse: 5′-GACTCCACGACGTACTCAGC-3′) was used as the endogenous housekeeping control gene. Each PCR reaction was performed in a 10 μL mixture containing 0.25 U GoTaq G2 Hot Start Polymerase (Promega), 1x GoTaq G2 Buffer, 1.5 mM MgCl$_2$ (Invitrogen), 200 nM of each primer (forward and reverse), 200 μM dNTP mixture containing the four deoxyribonucleotides (dATP, dCTP, dTTP, and dGTP), and 10 ng of cDNA template. Appropriate negative controls and genomic DNA positive controls were incorporated into each experiment. Amplification thermal program included an initial denaturation step of 95 °C for 3 min and 40 cycles of 95 °C for 15 s, 58 °C for 15 s and 72 °C for 15 s using the ProFlex$^{TM}$ PCR System Thermal Cycler (Applied Biosystems). Subsequently, total amount of PCR product was separated by electrophoresis at 120 V for 30 min in 2% agarose gel diluted in 1X Tris-acetate EDTA buffer (w/v) and stained with 0.01% of SYBR Safe (Thermo Fisher).

For real-time quantitative PCR, the reactions were conducted in three replicates with a final volume of 10 μL containing 1X GoTaq qPCR Master Mix (Promega), 300 nM CXR Reference Dye, 200 nM of each SYBR green designed primers: Angiotensin I Converting Enzyme 2 (ACE2; forward: 5′-CGAAGCCGAAGACCTGTTCTA-3′; reverse: 5′-GGGCAAGTGTGGACTGTTCC-3′); Natriuretic Peptide A (ANP; forward: 5′-CAACGCAGACCTGATGGATTT-3′; reverse: 5′-AGCCCCCGCTTCTTCATTC-3′); Spliced X-Box Binding Protein 1 (sXBP1; forward: 5′- CTGAGTCCGAATCAGGTGCAG-3′; reverse: 5′-ATCCATGGGGAGATGTTCTGG-3′); Unspliced X-Box Binding Protein 1 (usXBP1; forward: 5′-CAGCACTCAGAC TACGTGCA-3′; reverse: 5′-ATCCATGGGGAGATGTTCTGG-3′); Total X-Box Binding Protein 1 (totalXBP1; forward: 5′- TGGCCGGGTCTGCTGAGTCCG-3′; reverse: 5′-ATCCATGGGGAGATGTTCTGG-3′); C/EBP-homologous protein (CHOP; forward: 5′-AGAACCAGGAAACGGAAACAGA-3′; reverse: 5′-TCTCCTTCATGCGCTGCTTT-3′); Actinin Alpha 1 (ACTN1; forward: 5′-CCCGAGCTGATTGACTACGG-3′; reverse: 5′-GCAGTTCCAACGATGTCTTCG-3′); Actinin Alpha 2 (ACTN2; forward: 5′-GACATCGTGAACACCCCTAAAC-3′; reverse: 5′- CCGCAAAAGCGTGGTAGAA); Actin Alpha 1 (ACTA1; forward: 5′-TGCCAACAACGTCATGTCG-3′; reverse: 5′-CAGCGCGGTGATCTCTTTCT-3′); Troponin I3, Cardiac Type (TNNI3; forward: 5′-TTTGACCTTCGAGGCAAGTTT-3′; reverse: 5′-CCCGGTTTTCCTTCTCGGTG-3′); Troponin I1, Slow Skeletal Type (TNNT1; forward: 5′-TGATCCCGCCAAAGATCCC-3′; reverse: 5′-TCTTCCGCTGCTCGAAATGTA-3′); Myosin Heavy Chain 6 (MYH6; forward: 5′- GCCCTTTGACATTCGCACTG-3′; reverse:

5′-GGTTTCAGCAATGACCTTGCC-3′); Myosin Heavy Chain 7 (MYH7; forward: 5′-TCACCAACAACCCCTACGATT-3′; reverse: 5′- CTCCTCAGCGTCATCAATGGA-3′); and 10 ng of cDNA per reaction. The reactions were performed on a StepOnePlus ™ Real-Time PCR System thermocycler (Applied Biosystems). The relative expression of the genes of interest (GOI) was normalized by endogenous control genes: Glyceraldehyde-3-phosphate Dehydrogenase (GAPDH; forward: 5′-GCCCTCAACGACCACTTTG-3′; reverse: 5′-CCACCACCCTGTTGCTGTAG-3′) and Hypoxanthine Phosphoribosyl transferase 1 (HPRT-1; forward 5′-CGTCGTGATTAGTGATGATGAACC-3′; reverse: 5′-AGAGGGCTACAATGTGATGGC-3′).

Data analysis was performed with the $N_0$ method implemented in LinRegPCR *v.* 2020.0 software, which considers efficiency estimated by the window-of-linearity method, as proposed by *Ramakers et al. (2003)*, *Ruijter et al. (2009)*. $N_0$ values were calculated using default parameters and the arithmetic mean of $N_0$ values from GOI were normalized by taking its ratio to the $N_0$ of the geometric mean of the endogenous control genes (REF) GAPDH and HRRT-1 ($N_0$GOI /$N_0$REF).

## Protein expression

Media was completely removed from hiPSC-CMs, and 40 µL of sample buffer without bromophenol blue (62.5 mM Tris–HCl, pH 6.8, containing 10% glycerol, 2% SDS, and 5% 2-mercaptoethanol) was added to each well. Next, cell extracts were boiled at 95 °C for 5 min, centrifuged 16,000 x g for 15 min at 4 °C, and the supernatant was collected. Protein content was estimated using the Bio-Rad Protein Assay (#5000006; Bio-Rad). After the addition of bromophenol blue (0.02%), samples were separated by electrophoresis on an 8% SDS polyacrylamide gel and transferred to polyvinylidene difluoride (PVDF) membranes.

Western blotting was carried out with minor modifications from the described (*Towbin, Gordon & Staehelin, 1989*). Briefly, membranes were blocked in 5% non-fat milk diluted in Tris-Buffered Saline with 0.1% Tween-20 (TBS-T) for 1 h at room temperature. Membranes were then incubated overnight at 4 °C with primary antibodies diluted in TBS-T with 5% non-fat milk. Then, membranes were washed and incubated with peroxidase-conjugated antibodies. The signals were developed using ECL Prime Western Blotting System (# GERPN2232, Sigma) for 5 min and chemiluminescence was detected with an Odyssey-FC System® (LI-COR Biosciences).

Stripping protocol was performed to break bonds between the antibodies and the transferred proteins in order to reuse membranes. Briefly, membranes were incubated for three cycles of 10 min in a stripping buffer (pH 2.2, 200 mM glycine, SDS 0.1% and 1% Tween-20). Then, the buffer was discarded, the membranes were washed for 5 min with PBS and 5 min with 0.1% TBS-T (three times each). Next, membranes were blocked again and proceeded with the above-described steps. Full-length membranes are displayed in Fig. S5.

## Beating frequency evaluation

hiPSC-CMs were plated on 96-well plates (1.8 $\times 10^4$ cells per well) and treated with NE-100 1 µM or vehicle (DMSO) before incubating at 37 °C. At 24- and 48-hours post-treatment,

beating frequency was measured by manually counting the synchronous contractions of the monolayer for 60 s (beats per minute). Cells were observed using an EVOS cell imaging system (Thermo Fisher Scientific), in brightfield mode. Representative videos were recorded using Operetta® High-Content Screening System (Perkin Elmer) for each condition at baseline and 48 h after treatment with vehicle or NE-100.

## Statistical analysis

Data are presented as mean values, and error bars indicate the standard error of the mean (S.E.M). Statistical differences were analyzed using nested $t$-test, unpaired two-tailed Welch's $t$-test and between three or more groups, unpaired multiple t-tests (Holm-Sidak method) or ordinary one-way ANOVA with Holm-Sidak post-hoc. Prism v8.0 (GraphPad) was used for data analysis and graphics, where statistical significance was accepted at $P < 0.05$. The tests and $p$ values are specified in figure legends and the symbols represent $p < 0.05$, ** $p < 0.01$, * $p < 0.001$, *** $p < 0.0001$.

# RESULTS

## Human induced pluripotent stem cell-derived cardiomyocytes (hiPSC-CMs) express the Sigma-1 receptor (S1R)

Human cardiomyocytes were differentiated from induced pluripotent stem cells according to established protocols (*Cruvinel et al., 2020*). Differentiation to cardiomyocytes resulted in low expression of the pluripotency marker OCT-4 ($0.8\% \pm 0.4\%$) and high expression of the specific cardiac muscle marker troponin T (cTnT) ($88.4\% \pm 8.4\%$) (Fig. S1A). The presence of cTnT was confirmed by immunostaining, as was cell morphology with F-actin staining (Fig. S1B).

Previous studies showed that S1R was expressed in the human heart tissue and in atrial and ventricular cardiomyocytes of rodents (*Novakova et al., 1995*; *Novakova et al., 2010*; *Fagerberg et al., 2014*; *Abdullah et al., 2020*). Here, we showed S1R mRNA and protein in hiPSC-CMs (Figs. 1A and 1B, respectively). Immunostaining showed that S1R is in the nuclei and in the perinuclear region of the cells (Fig. 1C). This is consistent with previous findings of the distribution of S1R in cells from rodents and yeasts and in human cell lines (*Dussossoy et al., 1999*; *Hayashi & Su, 2003a*). S1R is enriched at the mitochondria-associated ER membrane (MAM) (*Hayashi & Su, 2007*; *Morales-Lázaro, González-Ramírez & Rosenbaum, 2019*). We found that part of cytosolic S1R staining was in close apposition with the mitochondrial marker VDAC1/Porin (Fig. 1D) and with the ER chaperone lectin Calnexin (CNX) (Fig. 1E) in hiPSC-CMs.

## Inhibition of S1R reduces infection and replication of SARS-CoV-2 and prevents cell death in human cardiomyocytes

To test whether S1R had a direct influence on SARS-CoV-2 infection in human cardiomyocytes, the selective S1R antagonist, NE-100, was used to inhibit this receptor in hiPSC-CMs prior to exposure to the virus. NE-100 has a binding affinity for S1R within the nanomolar range and an IC50 200 times higher for other receptors, such as dopamine, serotonin, and phencyclidine receptors (*Okuyama et al., 1993*). Exposure of

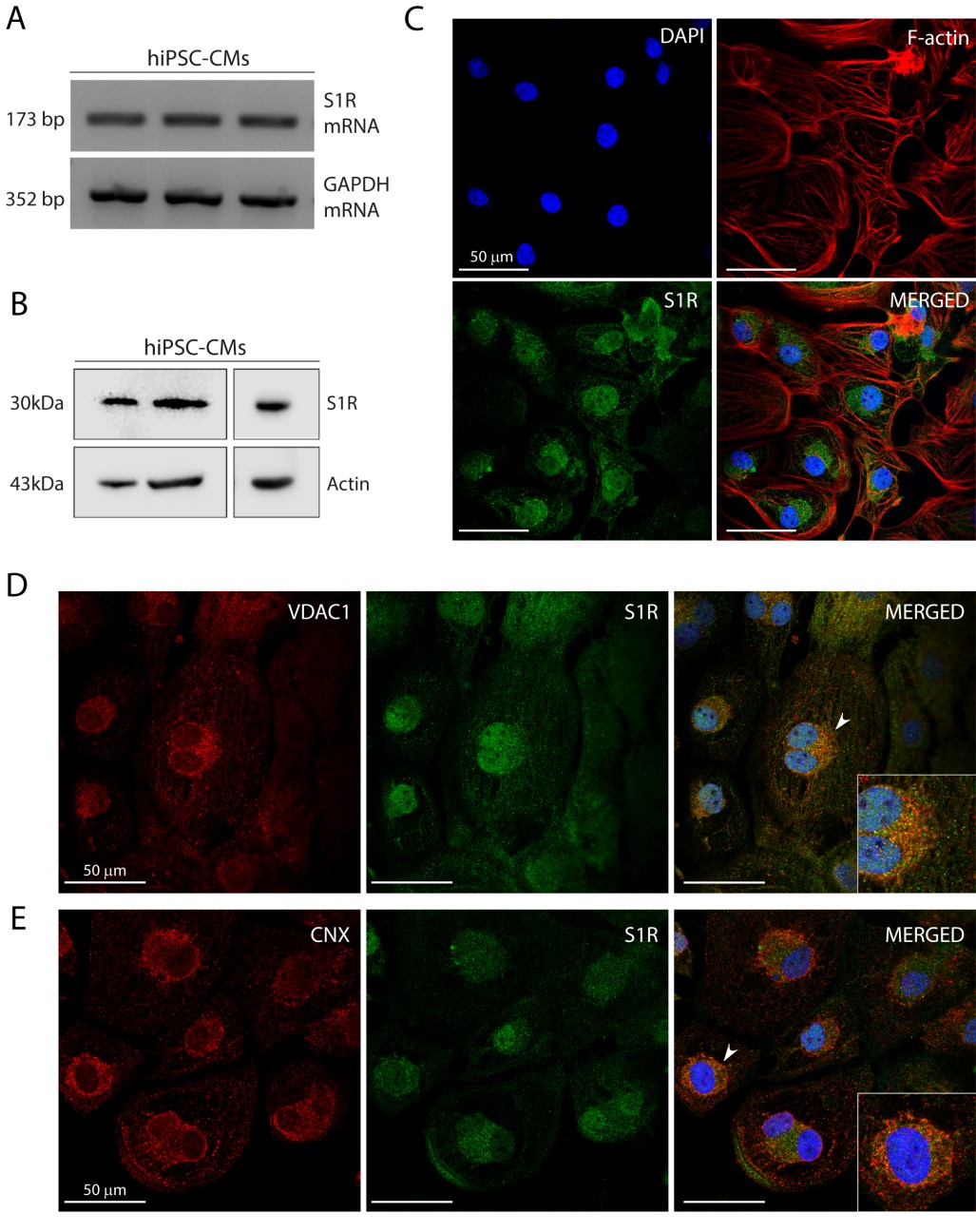

**Figure 1** **Human induced pluripotent stem cell-derived cardiomyocytes (hiPSC-CMs) express the Sigma-1 receptor (S1R).** (A) End point PCR analysis of S1R on mRNA isolated from three hiPSC-CMs replicate samples from three independent batches of differentiation ($N = 3$). GAPDH was used as housekeeping control of mRNA samples. Uncropped agarose gel is appended in Fig. S5 with appropriate controls. (B) Representative western blot of S1R expression in protein extracts of control hiPSC-CMs. Actin was used as a loading control. The detection of the protein was performed in samples from 3 different batches with similar results ($N = 3$). Full-length gels of the representative blots shown here are presented in Fig. S5. (continued on next page...)

**Figure 1 (...continued)**
(C) Representative image of staining performed in fixed cardiomyocytes from three different batches shows S1R presence and similar distribution in hiPSC-CMs ($N = 3$). S1R (green); phalloidin (red) and nuclei (blue); scale bar $= 50\,\mu$m. (D) Immunofluorescence shows mitochondrial marker VDAC1/Porin (red) and S1R (green) in hiPSC-CMs. Representative image of three independent experiments ($N = 3$). Nuclei were stained with DAPI (blue); scale bar $= 50\,\mu$m. (E) Immunofluorescence shows ER chaperone lectin Calnexin (CNX) (red) and S1R (green) in hiPSC-CMs. Representative image of 3 independent experiments ($N = 3$). Nuclei were stained with DAPI (blue); scale bar $= 50\,\mu$m. Zoom-in highlights the cells indicated by the arrowheads (D and E).

cardiomyocytes to NE-100 in the micromolar range for 24 h was reported to abolish the effects of a S1R agonist, similarly to what was observed using small interference RNA (siRNA) (*Tagashira et al., 2013*). Then, following 24 h exposure to 1 µM NE-100, hiPSC-CMs were infected with SARS-CoV-2 at the multiplicity of infection (MOI) 0.1. Immunofluorescence using convalescent serum (CS) from a recovered COVID-19 patient showed that 57.3% (±11.1%) of cells were infected at 48 h.p.i (hours post-infection) (Fig. 2A). CS-staining was confirmed with co-localization with the specific SARS-CoV-2 Spike Protein (SP) (Fig. S2).

Importantly, the inhibition of S1R reduced the percentage of infected hiPSC-CMs to 35.8% (±2.5%) (Figs. 2A and 2B). Moreover, whereas infection of hiPSC-CMs with SARS-CoV-2 led to the production of infectious virions measured in the plaque assay, exposure to NE-100 considerably diminished this viral yield, with an average reduction of 82% plaque forming units (PFU) at 48 h.p.i (Fig. 2C). SARS-CoV-2 infection was previously shown to induce cytopathic features in hiPSC-CMs, mostly related to myofibrillar disruption, as described by other groups (*Perez-Bermejo et al., 2021*; *Marchiano et al., 2021*). Likewise, we detected a pattern of structural disarrangement of cardiac troponin T staining, as shown in Fig. S3.

The release of lactate dehydrogenase (LDH) confirmed that SARS-CoV-2 caused cardiomyocyte death, as previously reported (*Sharma et al., 2020*). At 24, 48, and 72 h.p.i, LDH was elevated by 2.3, 5.6 and 2.7-fold, respectively, in SARS-CoV-2-infected cells when compared to controls (Fig. 2D). NE-100 was able to decrease the release of LDH at 48 h.p.i and also tended to decrease cytotoxicity at 72 h.p.i (Fig. 2D). There was no increase in the release of LDH when comparing uninfected to NE-100 treated controls (Fig. 2D). These results suggest that the inhibition of S1R does not induce cell death and may protect hiPSC-CMs by decreasing infection.

S1R is engaged in ER protein synthesis and acts as a chaperone for proteins translocating to cell surface (*Vela, 2020*). We investigated whether the inhibition of S1R could be related to modifications in the host cell receptor for viral entry, since it partially prevented the infection. The expression of the main viral entry receptor, ACE2, was measured after 24 h exposure to 1 µM NE-100. There was a reduction in the levels of ACE2 mRNA, albeit not statistically significant ($p = 0.0790$) (Fig. S4A). However, western blot analysis of ACE2 after NE-100 treatment did not show differences at the protein level (Figs. S4B and S4C). These data suggest that the inhibition of SARS-CoV-2 infection in hiPSC-CMs can

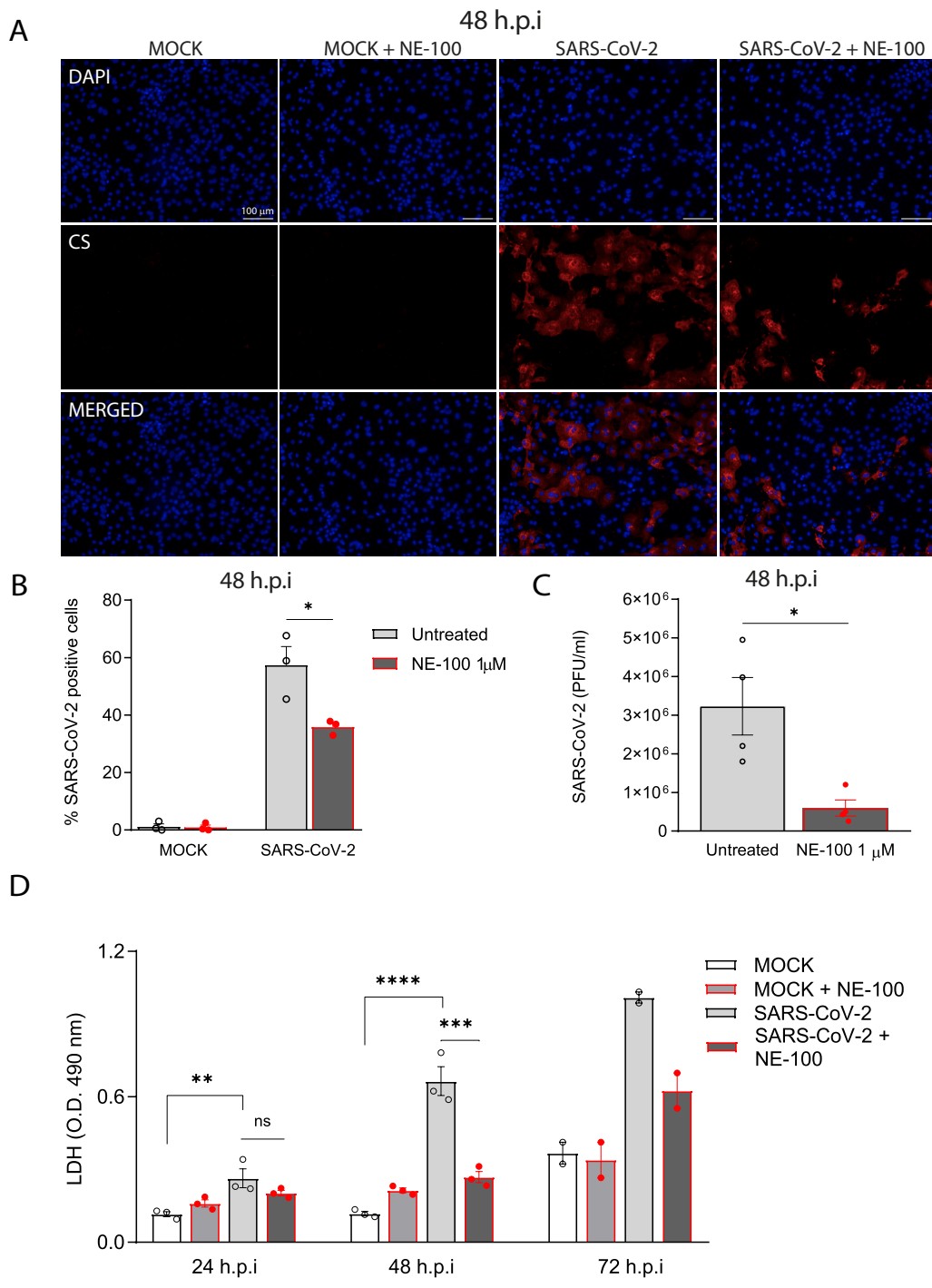

**Figure 2 S1R inhibition reduces SARS-CoV-2 infection, replication, and cytotoxicity in hiPSC-CMs.**
(A) hiPSC-CMs were pre-treated for 24 h with 1 μM NE-100 and infected with SARS-CoV-2 at multiplicity of infection (MOI) of 0.1. Cells were evaluated at 48 h post-infection (h.p.i). Representative immunostaining images show infected hiPSC-CMs positively stained for convalescent serum (CS) in red and no signal detection in mock conditions ($N = 3$); scale bar = 100 μm. 

**Figure 2 (...continued)**
(B) The percentage of infected hiPSC-CMs was assessed by quantification of CS positive cells in SARS-CoV-2-infected and mock-infected conditions exposed or not to S1R high-affinity antagonist NE-100 ($N = 3$). (C) Plaque forming units assay for the supernatants of the SARS-CoV-2 infected hiPSC-CMs ($N = 4$). (D) Cell death was measured in the supernatant by LDH activity at 24, 48 ($N = 3$) and 72 h.p.i ($N = 2$). Data are represented as the mean ± S.E.M, analyzed by Nested $t$-test ($p = 0.0003$) (B), unpaired two-tailed Welch's $t$-test ($p = 0.0336$) (C) and ordinary one-way ANOVA followed by Holm-Sidak's post-hoc (24 h $p = 0.0089$; 48h $p < 0.0001$ and $p = 0.0001$) (D). Data points represent independent experiments. * $p < 0.05$; ** $p < 0.01$; *** $p < 0.001$; **** $p < 0.0001$ $p < 0.0001$.

occur through mechanisms other than a reduction in the availability of ACE2 under the above-described conditions.

## NE-100 attenuates cytokine release by SARS-CoV-2 infected hiPSC-CMs

Cell death is the final denouement of hiPSC-CMs at 72 h.p.i (*Sharma et al., 2020*; *Aragão et al., 2021*). To investigate events before cell death, we analyzed some cytokines associated with COVID-19 at 24 and 48 h.p.i. This strategy was previously shown to be significant in the context of the infection of hiPSC-CMs by other pathogens that target the heart (*Bozzi et al., 2019*).

SARS-CoV-2 infection increased the release of interleukin (IL)-6 when compared to control. NE-100 decreased this response at 24 h.p.i and 48 h.p.i (Figs. 3A and 3B). Conversely, no differences in the secretion of IL-8 were observed at 24 h.p.i (Fig. 3A), however, at 48 h.p.i, SARS-CoV-2 increased IL-8 by 5.5-fold relative to the mock condition (Fig. 3B). Comparably to the modulation of IL-6, NE-100 decreased the secretion of IL-8 (Fig. 3B). Tumor necrosis factor-alpha (TNF-$\alpha$) levels were only transiently increased at 24 h.p.i (3.3-fold) but were nevertheless inhibited by previous NE-100 treatment (Fig. 3B).

## S1R inhibition decreases beating in cardiomyocytes

To investigate the consequences of the inhibition of S1R in human cardiomyocytes by itself, we conducted further tests on the effects of NE-100 in hiPSC-CMs. We first analyzed whether NE-100 induces cell death. NE-100 does not decrease cell viability after 72 h at concentrations ranging from 10 nM to 10 $\mu$M as measured by the neutral red assay (Fig. 4A). Also, hiPSC-CMs treated with 1 $\mu$M NE-100 for 48 h showed no changes in the number of pyknotic nuclei, which represents an irreversible chromatin-condensed nuclear state characteristic of cell death (Fig. 4B).

Furthermore, given that S1R function is important for the ER stress response, we investigated whether NE-100 modulates the endonuclease activity of inositol-requiring enzyme 1$\alpha$ (IRE1$\alpha$) by analysis of splicing of XBP1 mRNA, measuring the levels of spliced XBP1 mRNA (sXBP1), in addition to the unspliced (uXBP1) and total XBP1 mRNAs (*Oslowski & Urano, 2011*). We found low levels of sXBP1 mRNA and high levels of uXBP1 mRNA in hiPSC-CMs which are consistent with cells under non-ER stress condition (Fig. 4C). NE-100 treatment did not affect levels of sXBP1 or uXBP1 mRNAs/total XBP1 (Fig. 4C). These results indicate that NE-100 did not induce activation of IRE1$\alpha$ in hiPSC-CMs under these conditions. Another indicator of ER stress response is the upregulation

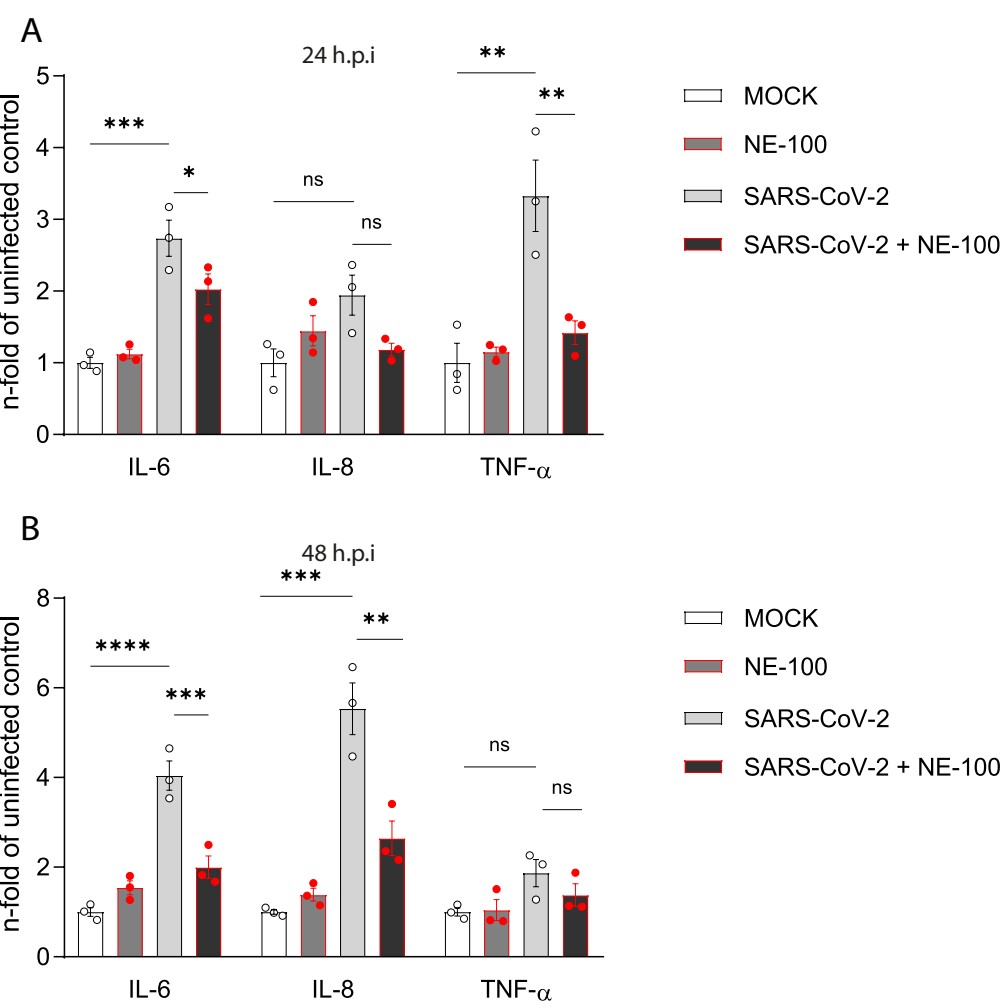

**Figure 3** **NE-100 decreases cytokine release that follows SARS-CoV-2 infection in hiPSC-CMs.** (A and B) hiPSC- hiPSC-CMs were pre-treated or not with NE-100 1 $\mu$M and infected with SARS-CoV-2. Supernatants were analyzed by ELISA for IL-6, IL-8 and TNF-$\alpha$ at 24 h.p.i (IL-6 $p = 0.0006$ and $p = 0.0395$; IL-8 $p = 0.0661$ and $p = 0.1426$; TNF-$\alpha$ $p = 0.0033$ and $p = 0.0077$) (A) and 48 h.p.i (IL-6 $p < 0.0001$ and $p = 0.0009$; IL-8 $p = 0.0001$ and $p = 0.0017$; TNF-$\alpha$ $p = 0.1663$ and $p = 0.5359$) (B) ($N = 3$). Data were normalized to the mean control value of each experiment and represents the mean $\pm$ SEM of fold increase, analyzed by ordinary one-way ANOVA with Holm-Sidak's post-hoc. Data points represent independent experiments. * $p < 0.05$; ** $p < 0.01$; *** $p < 0.001$; **** $p < 0.0001$. $p < 0.0001$.

of the C/EBP-homologous protein (CHOP or GADD153). Again, we did not find changes in CHOP mRNA after NE-100 treatment (Fig. 4D).

These data suggest that NE-100 does not induce hiPSC-CMs death or ER stress response in the absence of additional stressor stimuli. However, one of the main functional features displayed by cultured cardiomyocytes is their ability to perform spontaneous contractions. In that matter, hiPSC-CMs are suitable to evaluate drug-induced changes in contractility (*Pointon et al., 2015*; *Niehoff et al., 2019*). We observed a decrease from 17.7 ($\pm$5.9) beats per minute (bpm) to 6.7 ($\pm$1.8) bpm and from 14.7 ($\pm$2.8) bpm to 4.1 ($\pm$0.8) bpm after

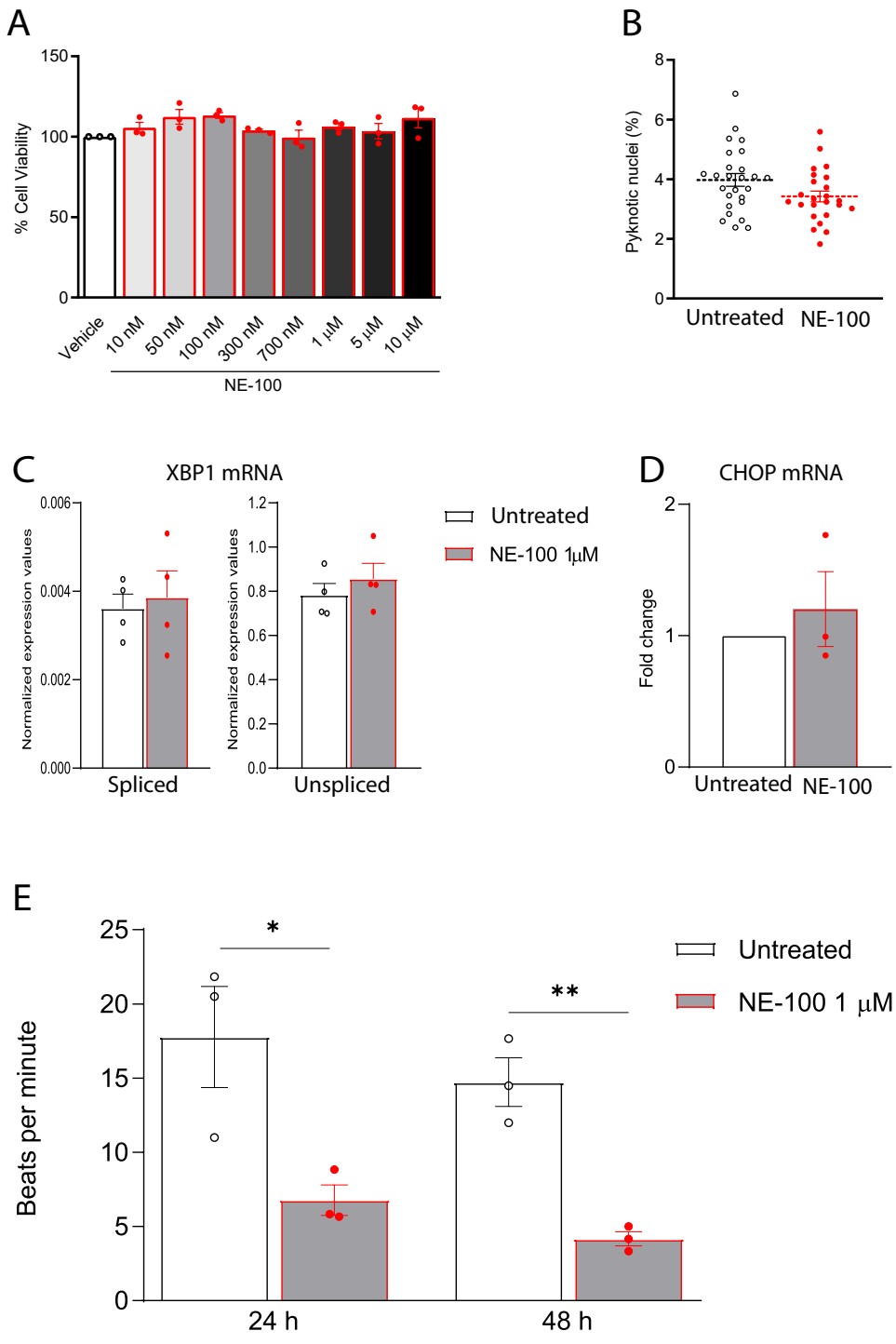

**Figure 4** **NE-100 does not induce death and ER stress but decreases beating frequency in hiPSC-CMs.**
(A) Neutral red cell viability assay for escalating NE-100 concentrations shows non-significant changes after 72 h post-treatment. Data are represented as the baseline-corrected mean ± S.E.M relative to the vehicle condition analyzed by one-way ANOVA ($N = 3$). (continued on next page...)

**Figure 4 (...continued)**
(B) Nuclear size analysis by DAPI staining shows the percentage of pyknotic nuclei after 48 h ($N = 3$); Dots represent the percentage of each well evaluated in three independent experiments analyzed by unpaired Welch's t test; non-significant ($p = 0.1738$). (C) Normalized mRNA expression of spliced and unspliced XBP1 transcripts in samples from different batches exposed to NE-100 1 μM in four independent experiments ($N = 4$). Data are expressed relative to total XBP1 normalized expression ± S.E.M and were analyzed by multiple t-tests (Holm-Sidak method) (spliced/total $p = 0.7338$ and unspliced/total $p = 0.6943$). (D) CHOP mRNA in samples from different batches exposed to NE-100 1 μM in three independent experiments ($N = 3$). Values are expressed as fold-change relative to the untreated condition ± S.E.M and were analyzed by unpaired Welch's t test; non-significant ($p = 0.5512$). (E) Average of beats per minute analyzed 24 and 48 h after exposure to NE-100 or vehicle ($N = 3$). Data are presented as the average ± S.E.M, statistical differences were analyzed by multiple t-tests (Holm-Sidak method) (24 h $p = 0.03663$; 48 h $p = 0.00349$).

24 h and 48 h of NE-100 exposure, respectively (Fig. 4E and Videos S1–S4). These results suggest that the inhibition of S1R leads to changes in beating rate and a putative deviation in contractility.

## The S1R antagonist NE-100 induces maladaptive transcriptional and structural changes in human cardiomyocytes

The findings on the decremental impact of NE-100 on the cardiomyocytes beating rate led us to pursue this issue further. S1R plays a cardioprotective role during maladaptive cardiac remodeling and the anti-hypertrophic properties of S1R agonists have been described (*Tagashira & Fukunaga, 2012*; *Tagashira et al., 2013*; *Hirano, Tagashira & Fukunaga, 2014*). Therefore, we also investigated if the S1R antagonist NE-100 alone could induce a hypertrophic response. To that end, we quantified the expression of key genes upon S1R inhibition. First, by measuring mRNA levels of the atrial natriuretic peptide (ANP), which is one of the main transcripts of the fetal program for cardiac growth. ANP overexpression is frequently correlated to cardiac hypertrophy but NE-100 decreased ANP mRNA expression by 1.49-fold after 24 h (Fig. 5A). There was also no increase in the average cell surface area of hiPSC-CMs observed by F-actin staining. In fact, more like the contrary, NE-100 exposure caused a decrease in the average cell surface area after 48 h (Fig. 5B), which is a phenotype opposite to the expected for cardiomyocyte hypertrophic responses *in vitro* (*Watkins et al., 2012*).

After observing cell shrinking in longer exposures to NE-100, we then hypothesized that S1R inhibition could be triggering morphological changes. Hence, we investigated the expression of myofibril-associated genes important for cytoarchitecture maintenance and observed a downregulation of 4 out of 7 genes investigated. The decreased expression of *ACTN2* (−1.53-fold), *ACTA1* (−11.1-fold), *TNNI3* (−2.32-fold) and *TNNT1* (−2.43-fold) after the inhibition of S1R suggests a transcriptional regulation that could impair sarcomeric organization and cardiomyocyte cytoskeletal integrity (Fig. 5C). *ACTN1* and *MYH7* expression did not show significant changes upon NE-100 exposure (Fig. 5C). Interestingly, *MYH7* upregulation is highly associated with hypertrophy (*Krenz & Robbins, 2004*; *Gupta, 2007*). Therefore, this result corroborates the assumption that the inhibition of S1R in the absence of a hypertrophic stimulus is not sufficient to activate the molecular profile of cardiac hypertrophy in hiPSC-CMs (*i.e., ANP* and *MYH7* upregulation). Intriguingly,

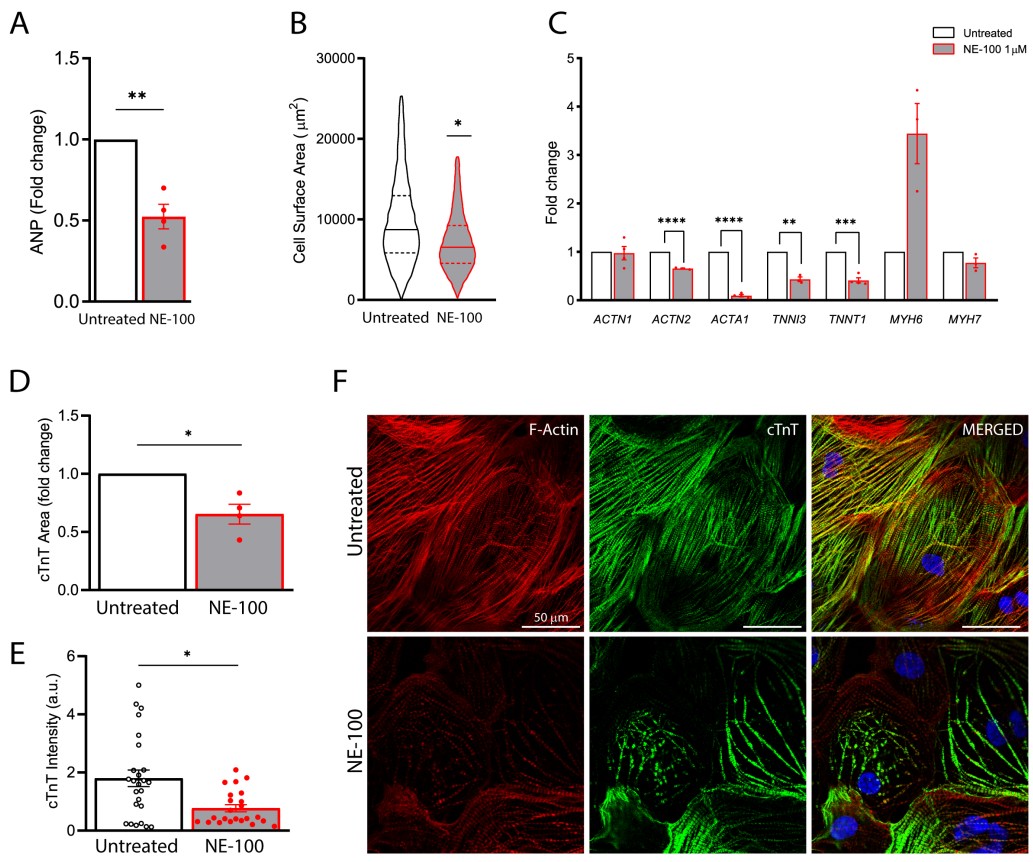

**Figure 5  S1R inhibition downregulates the expression of structural-related genes and compromise cytoskeletal integrity.** (A) Real-time qPCR shows decreased levels of transcript content for ANP after S1R inhibition ($N = 4$). (B) Cell area was quantified by F-actin staining and shows a significant decrease in cell body sizes after exposure to NE-100 for 48 h. Data represents distribution of surface area measured from approximately 24000 cells in four independent experiments ($N = 4$). (C) Changes in myofibril and cytoskeletal-related gene expression after 24 h of NE-100 1 μM exposure from at least three replicates obtained from independent experiments ($N = 3$ or $N = 4$). (D and E) Quantification of cTnT immunoreactive area and intensity, normalized by the total number of cells per field; values are expressed relative to untreated controls (D) or arbitrary units, and data points represent mean cTnT intensity per well (E) ($N = 4$). (F) Representative confocal images show in more detail the disruption of F-actin and cTnT organization, observed in at least four independent repeats ($N = 4$). Scale bar = 50 μm. Data are presented as the average ± S.E.M, statistical differences were analyzed by unpaired Welch's $t$-test ($p = 0.0081$ and $p = 0.0262$) (A and D), nested $t$-test ($p = 0.0108$) (B), multiple t-tests (Holm-Sidak method) (ACTN1 $p = 0.8478$, ACTN2 $p = 0.000004$, ACTA1 $p < 0.000001$, TNNI3 $p = 0.0011$, TNNT1 $p = 0.0001$, MYH6 $p = 0.0502$, MYH7 $p = 0.1582$) (C), and Mann–Whitney test ($p = 0.0235$) (E). Data points represent independent experiments unless otherwise stated. * $p < 0.05$; ** $p < 0.01$; *** $p < 0.001$; **** $p < 0.0001$.

*MYH6* was the only transcript upregulated by 3.44-fold ($p = 0.0502$) (Fig. 5C), which could represent a transcriptional compensatory adaptation.

Furthermore, cardiac troponin T (cTnT) staining analysis revealed a significant decrease both in immunoreactive area and fluorescence intensity on hiPSC-CMs exposed to NE-100 when compared to untreated controls (Figs. 5D and 5E, respectively). These results are probably attributable also to a decrease in cellular cTnT content rather than exclusively

to a change in the cell area, indicating a considerable loss of troponin. Consistently, the morphology of cytoskeletal fibers was grossly affected after 48 h, as shown in confocal fluorescence images (Fig. 5F). Taken together, our results suggest that S1R inhibition caused neither change in cell viability nor induction of hypertrophy parameters. However, despite this apparent lack of negative effects related to cell death and hypertrophy, there was a downregulation of genes encoding structural proteins and cytoskeletal impairment *via* a decrease in cardiac troponin T content. The latter significant effects could explain in part the beating frequency reduction of human cardiomyocytes exposed to NE-100.

## DISCUSSION

In this study, we show that hiPSC-CMs are permissive to productive infection of SARS-CoV-2, corroborating previous reports (*Pietro et al., 2020*; *Sharma et al., 2020*; *Choi et al., 2020*; *Bojkova et al., 2020*; *Bailey et al., 2021*; *Perez-Bermejo et al., 2021*; *Marchiano et al., 2021*). Although the presence of SARS-CoV-2 in the postmortem heart is still controversial, the cytopathic features following cardiomyocyte infection *in vitro* are similar to those observed in the cardiac tissue of patients deceased from COVID-19, including sarcomeric fragmentation, innate immune cells activation and cell death (*Lindner et al., 2020*; *Bailey et al., 2021*; *Perez-Bermejo et al., 2021*). Here, we confirmed that SARS-CoV-2 induces cardiotoxicity by directly infecting cardiomyocytes.

Aside from viral infection and replication, immune responses are a hallmark of COVID-19 pathology that can also be modeled using hiPSC-CMs (*Azkur et al., 2020*; *Dariolli et al., 2021*). Along with monocytes and fibroblasts, cardiomyocytes are considered an important source of cytokines during events such as heart failure (*Aoyagi & Matsui, 2012*). In fact, Wong and collaborators have shown that SARS-CoV-2 upregulates the expression of pro-inflammatory cytokines in hiPSC-CMs, including IL-6, IL-8, and TNF-α (*Wong et al., 2020*). Here, we confirmed the release of cytokines after the infection with SARS-CoV-2. Elevated levels of IL-6 are correlated with cardiac damage and heart failure in rodent models (*Janssen et al., 2005*). Moreover, previous reports demonstrated that IL-6 produced by cardiomyocytes promotes inflammation in the heart by recruiting neutrophils (*Youker et al., 1992*). Interestingly, IL-8, which was also increased, is a neutrophil chemotactic factor correlated with heart disease (*Rothenbacher et al., 2006*; *Akasaka et al., 2006*). Myocardial infiltration of neutrophils has been reported in the hearts of COVID-19 patients and is associated with myocardial damage (*Yao et al., 2020*).

Nonetheless, the characterization of the interlinkage between the virus and intracellular structures may provide therapeutic clues against COVID-19. A SARS-CoV-2–human protein interaction map showed that NSP6 interacts directly with S1R (*Gordon et al., 2020b*). The antiviral activity of sigma receptor ligands was described in Vero E6 cells (*Gordon et al., 2020b*). S1R knockdown in A549 cells and S1R knockout in Caco-2 cells further confirmed S1R, but not Sigma-2 receptor, as a host-dependency factor for SARS-CoV-2 infection (*Gordon et al., 2020a*). NSP6 orchestrates vesicle trafficking and regulates ER remodeling during the replication of mammalian coronaviruses (*Cottam et al., 2011b*) and its interaction with S1R possibly contributes to the rearrangement of endomembrane

compartments. S1R is critical for the replication of Hepatitis C Virus (HCV) that occur inside ER, where this receptor is mostly expressed (*Ritzenthaler & Elamawi, 2006*; *Friesland et al., 2013*). Interestingly, HCV is a positive-sense single-stranded RNA virus. Altogether, these data support S1R as a host target to dampen SARS-CoV-2 replication. On the other hand, the function of S1R in the homeostasis of cardiac cells makes this receptor a target regarding cellular impairments. Also, since COVID-19 triggers cardiac dysfunctions, drugs with negative cardiac effects would be unfeasible as a therapeutic approach (*Wu et al., 2020*; *Magadum & Kishore, 2020*). The inhibition of S1R could be detrimental to cardiac physiology since this receptor is important for bioenergetics and response to stress in heart cells (*Shinoda et al., 2016a*; *Shinoda et al., 2016b*; *Stracina & Novakova, 2018*).

The inhibition of S1R with NE-100 reduced the number of infected cells, the replication of SARS-CoV-2, prevented infection-associated cell death and attenuated the release of cytokines. However, it is the activation of S1R that has been associated with anti-inflammatory properties (*Szabo et al., 2014*; *Rosen et al., 2019*; *Zhou et al., 2019*). It was recently reported that the S1R agonist fluvoxamine prevents clinical deterioration of symptomatic COVID-19 and the persistence of residual symptoms (*Lenze et al., 2020*; *Seftel & Boulware, 2021*). Consequently, the reduction of pro-inflammatory cytokines described here is more likely due to the inhibition of infection rather than a direct suppression of the release of cytokines.

Although S1R is foregrounded in COVID-19 research, it is currently unclear if a S1R antagonist or agonist could be better to treat the disease, since *in vitro* reports show that S1R antagonists prevent SARS-CoV-2 replication and S1R agonists have proviral activity, while clinical evidence suggests that S1R agonists prevent worsening of symptoms. In spite of that, evidence of side effects of S1R ligands, especially cardiac-specific detrimental cellular effects, are overlooked in current studies and should be acknowledged for further treatment options. In order to look into the consequences of S1R inhibition *per se* on hiPSC-CMs, other aspects were investigated beyond antiviral activity. The survival of cardiomyocytes was not affected by NE-100. Given the role of S1R in ER stress response, the mRNA levels of sXBP1 and CHOP were investigated as another aspect of toxicity (*Wang et al., 2018*). The spliced XBP1 analysis represents a reliable indirect method to determine IRE1α activation, which is one of the major ER stress sensors involved in pro-inflammatory cytokine production (*Oslowski & Urano, 2011*; *Rosen et al., 2019*).

Alam and collaborators' have demonstrated that the knockdown of S1R increases the expression of CHOP in neonatal rat ventricular cardiomyocytes, while decreases spliced XBP1 with tunicamycin (*Alam et al., 2017*). It means that S1R is crucial for coping with ER stress and that its depletion favors cytotoxic ER stress response in cardiac cells by CHOP upregulation and impaired sXBP1-mediated signaling. In the present study, no changes in the transcript levels of sXBP1 and CHOP were detected in hiPSC-CMs with NE-100. Our results suggest that there is no intrinsic ER stress induction by NE-100, but future studies should explore the role of S1R ligands in ER stress transcripts' expression upon ER-stressor stimuli in hiPSC-CMs.

Nonetheless, we observed that inhibition of S1R reduced the frequency of cardiomyocytes' beating. Consistent with our findings, it has been noted that S1R
antagonists cause delayed cardiac repolarization, impairment of rate adaptation, and increased risk for drug-induced arrhythmia (*Witchel, 2011*; *Balasuriya et al., 2014*; *Morales-Lázaro, González-Ramírez & Rosenbaum, 2019*). Interestingly, haloperidol, which is a S1R antagonist, reduces the replication of SARS-CoV-2 (*Gordon et al., 2020b*; *Gordon et al., 2020a*). However, this antipsychotic did not decrease the risk for intubation or death in hospitalized patients (*Hoertel et al., 2021*). The anti-SARS-CoV-2 activity of the broad-spectrum antiviral remdesivir was described in human cardiomyocytes by Choi and colleagues, together with a safety profile evaluation that stipulated considerable arrhythmogenic and cardiotoxic risk *in vitro*. These results raised concerns regarding the drug-induced cardiotoxicity of repositioned pre-approved compounds to manage COVID-19 (*Choi et al., 2020*). The contractile capacity of cardiomyocytes is related to calcium availability. Since S1R is a key regulator of intracellular calcium homeostasis, the possibility that blocking its function may hamper the availability of cytoplasmic calcium and decrease beating rate in hiPSC-CMs should be considered. Indeed, previous studies showed an interconnection among S1R and mitochondrial function, calcium handling and contractility (*Ela et al., 1994*; *Tarabová, Nováková & Lacinová, 2009*). Although there is abundant data on S1R-dependent modulation of calcium homeostasis, ER stress response and hypertrophy (*Bhuiyan & Fukunaga, 2009*; *Tagashira & Fukunaga, 2012*; *Tagashira et al., 2013*), no previous study explored the consequences of the inhibition of S1R in the integrity of human cardiomyocytes' cytoskeleton.

S1R is a direct target of methamphetamine (*Nguyen et al., 2005*). Heart tissue from S1R knockout mice and autopsy samples of human methamphetamine users present fibrosis and signs of contractile dysfunction, suggesting a correlation between S1R and heart tissue remodeling (*Abdullah et al., 2018*; *Abdullah et al., 2020*). Although S1R activation is known to protect cardiomyocytes from hypertrophy upon pressure-overload, in the present study exposure of hiPSC-CMs to NE-100 with no secondary hypertrophic stimulus was not sufficient to neither increase the expression of *ANP* and *MYH7* nor cellular size.

The association between S1R and sarcomeric integrity was hypothesized (*Alam et al., 2017*), but morphological changes in human cardiomyocytes following S1R inhibition were not explored. Herein, S1R inhibition compromised the morphology of hiPSC-CMs with a perturbation of cytoskeletal architecture. The changes in morphology were accompanied by a decrease in cardiac troponin T and preceded by a downregulation of genes encoding key cytoskeletal proteins in cytoarchitecture maintenance, such as α-actinin 2, actin-α 1, troponin I3 and troponin T1. Importantly, the downregulation of these genes is associated with cardiotoxicity induced by doxorubicin, daunorubicin and mitoxantrone (*Chaudhari et al., 2016*). Out of seven targets investigated, *MYH6* was the only transcript increased with NE-100. It was previously demonstrated that the expression of myosin heavy chain is specifically induced as a compensatory response following disruption of the myofibrillar structure in hiPSC-CMs (*Perez-Bermejo et al., 2021*). Future studies should elucidate the mechanisms behind cytoskeletal changes and cardiomyocyte recovery after inhibition of S1R.

## CONCLUSIONS

The disruption of cytoskeletal and sarcomeric proteins by a decrease in expression or anomalous arrangement could underlie cardiomyopathies and heart failure, representing a potential mechanism by which drugs inhibiting S1R jeopardize cardiac integrity and contractility (*Sequeira et al., 2014*). Our results suggest that the inhibition of S1R leads to morphological and transcriptional changes in human cardiac cells and, therefore, its use as a therapeutic strategy against COVID-19 should be further investigated before translated to clinical application, due to the concern of drug-induced cardiac damage and malfunction.

## ACKNOWLEDGEMENTS

We thank Professor Leticia Raposo for the support in the statistical analysis of the data presented in this manuscript. We also thank Emiliano Horacio Medei and Renata Junqueira Moll Bernardes for scientific discussions. This work was developed as part of the doctoral thesis of J.A.S. at the Morphological Sciences Program, Biomedical Sciences Institute, Federal University of Rio de Janeiro (UFRJ), Rio de Janeiro, Brazil.

### Funding

This work was supported by intramural grants from D'Or Institute for Research and Education. Students' scholarships and fellowships were paid by the Coordenação de Aperfeiçoamento de Pessoal de Nível Superior (CAPES) or Conselho Nacional de Desenvolvimento Científico e Tecnológico (CNPq). The funders had no role in study design, data collection and analysis, decision to publish, or preparation of the manuscript.

### Grant Disclosures

The following grant information was disclosed by the authors:
D'Or Institute for Research and Education.
The Coordenação de Aperfeiçoamento de Pessoal de Nível Superior (CAPES) or Conselho Nacional de Desenvolvimento Científico e Tecnológico (CNPq).

### Competing Interests

Rafael Dariolli and Diogo Biagi were employed by PluriCell Biotech. The remaining authors declare that they have no competing interests.

### Author Contributions

- José Alexandre Salerno, Thayana Torquato and Karina Karmirian conceived and designed the experiments, performed the experiments, analyzed the data, prepared figures and/or tables, authored or reviewed drafts of the paper, and approved the final draft.
- Jairo R. Temerozo, Livia Goto-Silva and Mayara A. Mendes conceived and designed the experiments, performed the experiments, analyzed the data, authored or reviewed drafts of the paper, and approved the final draft.

- Carolina Q. Sacramento, Natalia Fintelman-Rodrigues, Letícia R.Q. Souza, Luiz Guilherme H.S. Aragão, Gabriela Vitória, Carolina S.G. Pedrosa, Suelen da Silva Gomes Dias and Vinicius Cardoso Soares performed the experiments, authored or reviewed drafts of the paper, and approved the final draft.
- Isis M. Ornelas and Carla P. Veríssimo and Teresa Puig-Pijuan performed the experiments, analyzed the data, authored or reviewed drafts of the paper, and approved the final draft.
- Vinícius Salazar performed the experiments, analyzed the data, authored or reviewed drafts of the paper, performed software programming and analysis, and approved the final draft.
- Rafael Dariolli, Diogo Biagi, Luciana Barreto Chiarini and Stevens K. Rehen conceived and designed the experiments, authored or reviewed drafts of the paper, provided resources, and approved the final draft.
- Daniel R. Furtado, Helena L. Borges, Patrícia T. Bozza, Marilia Zaluar P. Guimarães and Thiago M.L. Souza conceived and designed the experiments, authored or reviewed drafts of the paper, and approved the final draft.

## Data Availability

Full uncropped gels and membranes and the raw measurements for all results presented in each figure are available in the Supplemental Files.

## Supplemental Information

Supplemental information for this article can be found online at http://dx.doi.org/10.7717/peerj.12595#supplemental-information.

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
