# Peer review of "Inhibition of SARS-CoV-2 infection in human iPSC-derived cardiomyocytes by targeting the Sigma-1 receptor disrupts cytoarchitecture and beating"

_PeerJ, doi:10.7717/peerj.12595_

## Round 0.1 · original submission · Minor Revisions

Both reviewers are positive about this work. One has suggested some extra experiments with sigma-1 receptor agonists, which would be appropriate. While I would urge you to address this point, ideally with extra work, if you think this is beyond scope then please address this in your discussion.

Reviewer 1 ·

Basic reporting

The authors examined the effects of selective sigma-1 receptor antagonist NE-100 in iPS-derived human cadiomyocytes. Inhibition of sigma-1 receptor by NE-100 decreased both the number of infected cells and viral particles after 48 hours, and it prevented the release of pro-inflammatory cytokines and cell death. In contrast, NE-100 compromised cytoskeleton integrity by downregulating the expression of structural-related genes and reducing beating frequency.

Experimental design

Fresh human iPSCs-derived cardiomyocytes were used.

Validity of the findings

no comment

Additional comments

In this study, the authors used the selective sigma-1 receptor antagonist NE-100. However, sigma-1 receptor antagonist haloperidol did not show beneficial effects in COVID-19 patients. However, sigma-1 receptor agonists such as fluvoxamine and fluoxetine cause beneficial effects for COVID-19. From the current data, it is unclear whether antagonist or agonist show beneficial effects for COVID-19. Therefore, the additional experiments using the selective sigma-1 receptor agonist (i.e., cutamesine: SA4503) is needed. Cutamesine (SA4503) is available from commercially.

Reviewer 2 ·

Basic reporting

The paper is clearly written and the methodology detailed (including primer sequences and clone numbers for antibodies) and logical to follow. Permissions for use of clinical samples as well as the work at biosafety level 3 as per WHO guidelines are clearly stated. Results are clearly presented with individual data points in logically presented figures, including western blots (and original uncropped images) with molecular size markers and scale bars on photomicrographs and an excel file is provided containing raw data. Supplementary figures and videos provide further relevant information.

Experimental design

Experiments are well designed and the results presented clearly and logically to the review. Please address the comment below.

Specific comments
1. Statistical analysis should not be performed on Fig 2D 72h timepoint as the legend indicates n=2.

Validity of the findings

Some findings replicate previous reports, eg SARS-COV2 infection of hiPSC-CMs and the role of S1R (and its potential as a therapeutic target), the role of S1R in cardiomyocyte function and are therefore useful additional data and essentially the manuscript brings these areas of research together adding to previously reported knowledge but in a relevant cell model for assessing drug responses and drug toxicity. Overall, these findings would be useful to the research community

---

## Round 0.2 · accepted · Accept

Thank you for attending to the issues raised. I am satisfied with your rebuttal of the issue about S1R antagonism/agonism and your additional comment is perfectly judged. Similarly, the statistical issue has been resolved to my satisfaction so I see no need for further review by the external reviewers and I am happy to recommend acceptance.

Thanks for submitting this to PeerJ.